# siRNA Transfection Mediated by Chitosan Microparticles for the Treatment of HIV-1 Infection of Human Cell Lines

**DOI:** 10.3390/ma15155340

**Published:** 2022-08-03

**Authors:** Laura Chronopoulou, Francesca Falasca, Federica Di Fonzo, Ombretta Turriziani, Cleofe Palocci

**Affiliations:** 1Department of Chemistry, Sapienza University, 00185 Rome, Italy; laura.chronopoulou@uniroma1.it; 2Department of Molecular Medicine, Sapienza University, 00185 Rome, Italy; francesca.falasca@uniroma1.it (F.F.); ombretta.turriziani@uniroma1.it (O.T.); 3Department of Biochemical Sciences “Rossi Fanelli”, Sapienza University, 00185 Rome, Italy; federica.difonzo@uniroma1.it; 4CIABC-Centro di Ricerca per le Scienze Applicate alla Protezione dell’Ambiente e dei Beni Culturali, Sapienza University, 00185 Rome, Italy

**Keywords:** chitosan, polyplexes, gene delivery, HIV-1, siRNA, viral inhibition

## Abstract

Gene delivery is the basis for developing gene therapies that, in the future, may be able to cure virtually any disease, including viral infections. The use of short interfering RNAs (siRNAs) targeting viral replication is a novel strategy for treating HIV-1 infection. In this study, we prepared chitosan particles containing siRNA tat/rev via ionotropic gelation. Chitosan-based particles were efficiently internalized by cells, as evidenced by fluorescence microscopy. The antiviral effect of chitosan-based particles was studied on the C8166 cell line infected with HIV-1 and compared with the use of commercial liposomes (ESCORT). A significant reduction in HIV replication was also observed in cells treated with empty chitosan particles, suggesting that chitosan may interfere with the early steps of the HIV life cycle and have a synergic effect with siRNA to reduce viral replication significantly.

## 1. Introduction

One of the most promising trends in biomedicine that may revolutionize the way we treat diseases in the near future is the transition from the delivery of pharmaceuticals to the manipulation of gene expression [1,2,3,4]. As is well known, by using small interfering RNA-based molecules (siRNAs), it is possible to silence a specific gene and inhibit the associated protein [5]. This strategy has opened new doors for the treatment of a wide range of diseases [6,7]. However, efficient delivery of SiRNAs remains a challenge, especially due to their molecular instability in biological environments and their inability to easily cross cell membranes.

Moreover, RNA interference (RNAi) is a post-transcriptional gene silencing mechanism, highly conserved in eukaryotes, and it is a natural antiviral defense in plants, fungi and invertebrates [8,9]. In fact, viral infection induces the Dicer endoribonuclease-mediated production of virus-derived siRNAs (small interfering RNAs) that are transferred into Argonaute (AGO) proteins, the core components of RNA-induced silencing complex (RISC), possessing antiviral activity [10]. It was quite recently demonstrated that RNAi can effectively provide an antiviral immunity in mammals against a human virus (HEV71) [11].

The recent literature reported many RNAi-based therapeutic strategies that have the potential to treat a wide variety of diseases, including viral infections. This strategy could be particularly useful when a vaccine is not readily available or against new emerging viruses [12,13,14].

The major challenge that has to be overcome for a widespread therapeutic application of siRNAs is their delivery [15]. siRNAs have a relatively large molecular weight (~13 kDa) and are highly anionic, preventing their diffusion across cell membranes. Moreover, siRNAs are unstable in the bloodstream and can induce immune responses. Currently, different strategies for siRNA delivery are being developed [16]. Both viral and non-viral systems have been widely used for the delivery of genes and nucleic acids [17,18]. Non-viral delivery vectors are generally considered safer and more suitable alternatives to viral ones because of advantages such as easy synthesis, low immunogenicity, specific cell/tissue targeting and unlimited plasmid size [19,20,21,22]. Among them, chitosan (CS) and polyethylenimine (PEI) are considered the most promising cationic polymers to achieve a high efficient transfection through the formation of complexes with negatively charged nucleic acids. CS possesses many appealing physicochemical features such as biocompatibility, biodegradability, nontoxicity and non-immunogenicity and PEI, but on the other hand, it is well known to be the most effective polymer for gene/siRNA delivery due to its high proton-buffering capacity [23,24]. Moreover, the physiochemical properties of polymers can be tailored in order to lead to the formation of desired micro- and nanostructured assemblies, including block- and star-shaped copolymers, micelles, dendrimers, solid nanoparticles (NPs), polyplexes and polymer-siRNA conjugates [25,26,27,28,29]. The recent optimization of many technological approaches to synthetize biopolymeric-based carriers could offer precious tools and strategies for an efficient siRNA delivery.

In the present study, we aimed to design an efficient delivery system for siRNA to silence the HIV-1tat gene [30]. We employed CS-based particles with the aim of developing a simple and promising siRNA delivery system for the C8166 cell line. We finally evaluated the cytotoxicity and cellular uptake of the particles in mammalian cells as a function of the N:P ratio and chemical structure of particles, comparing polymeric-based particles (CS and PEI) and commercial liposomes (ESCORT).

## 2. Materials and Methods

### 2.1. Materials

Polyethylenimine (PEI, linear, MW 25,000 Da) and siRNA tat/rev (27 bp) were obtained from Polysciences Inc. (Warrington, PA, USA) and Bio-Fab Research (Rome, Italy), respectively. All other chemicals, including Chitosan (CS, 80% deacetylated, MW: 50 and 150 KDa), herring sperm DNA crude oligonucleotides < 50 bp (hsDNA) and fluorescein isothiocyanate (FITC) were from Sigma Aldrich (St. Louis, MO, USA) and used as received. Escort™ IV Transfection Reagent was purchased from Merck (Darmstadt, Germany). All solvents used were of analytical grade, purchased from Carlo Erba (Milan, Italy) and used as received.

### 2.2. Preparation and Characterization of CS-Based Complexes

Two commercial CS samples of different molecular weights (50 and 150 KDa) were used to prepare CS-based microparticles. CS was dissolved in 0.1 M acetic acid (pH 4.0) at 50 °C and then filtered through 0.22 µm Millex-GS filters (Millipore, Carrigtwohill, Ireland). To a 1 mg/mL CS solution, fixed amounts of a 1 mg/mL hsDNA solution (or 20 µM siRNA tat/rev solution) were added dropwise for electrostatic complexation between CS and nucleic acid. The preparation was stirred for 15 min, incubated for 30 min at RT and then stored at 4 °C for 24 h. Before each experiment, the suspension was sonicated for 30 min using a SONICA^®^ Ultrasonic Cleaner 2200 MH (Soltec s.r.l., Milan, Italy). Different ratios among CS aminic groups and hsDNA or siRNA phosphate groups (i.e., the N/P ratio) were studied, namely 5, 10 and 20, as reported in Table 1. Hydrodynamic size and Z-potential of microcomplexes were calculated by dynamic light scattering measurements (DLS) using a Nano ZetaSizer (Malvern Instruments Ltd., Malvern, UK). The CS/hsDNA complex morphology was characterized by field emission scanning electron microscopy using an Auriga 405 microscope (Carl Zeiss Microscopy GmbH, Oberkochen, Germany). One drop of complex solution was deposited on aluminum stabs and dried at RT. Subsequently, the sample was analyzed at 1.5–4 KeV.

CS/hsDNA samples, prepared as described above, were centrifuged (14,000 rpm, 20 min, 4 °C) with a D3024R microcentrifuge (SCILOGEX, Rocky Hill, CT, USA), and the amount of unbound hsDNA in the supernatant was determined spectrophotometrically by using a UV/Vis Ultrospec 4000 spectrophotometer (Pharmacia Biotech, Amersham, UK). The entrapment efficiency was calculated as follows:$$EE=\frac{initial\;amount\;of\;hsDNA-hsDNA\;amount\;in\;the\;supernatant}{initial\;amount\;of\;hsDNA}\;\;\times \;100$$

### 2.3. DNA Release Studies

For in vitro release kinetics studies, CS/hsDNA samples were prepared and freeze-dried. The samples were then incubated in 0.1 M phosphate buffer (pH 7.4) at a 1 mg/mL concentration and at 37 °C under magnetic stirring in a thermostatic bath (Intercontinental Equipment Inc., Kent, DE, USA). At fixed time intervals, samples were withdrawn and centrifuged (14,000 rpm, 20 min, 4 °C), and the amount of free DNA released in the supernatant was determined spectrophotometrically by comparing the absorbance of the samples with a calibration line.

### 2.4. Electrophoretic Mobility Shift Assay

The inclusion of hsDNA in CS-based NPs was monitored using agarose gel electrophoresis. CS/hsDNA nanocomplexes were loaded onto 2% (*w*/*v*) agarose gel (0.5 × TBE) containing ethidium bromide and electrophoresed at a constant voltage of 75 V for 30 min in 0.5 × TBE buffer. Chitosan/hsDNA samples were treated with DNase I and inactivated DNase I. The resulting hsDNA migration pattern was visualized under a UV transilluminator.

### 2.5. Preparation and Characterization of FITC-Labelled CS

The conjugation of CS with FITC was based on the reaction between the isothiocyanate group of FITC and the primary free amino group of CS. Equal volumes of CS 50 KDa 1 mg/mL in 0.1 M acetic acid solution and FITC 0.5 mg/mL methanolic solution were mixed (CS/FITC 2:1 *w*/*w* ratio) and incubated for 3 h in the dark at room temperature (20 °C). Then, labeled CS was precipitated through rising pH to 10 by adding 0.5 M NaOH. The mixture was centrifuged (14,000 rpm, 20 min, 4 °C) and washed with H_2_O. The FITC-CS was re-dissolved in 0.1 M acetic acid solution (pH 4) and dialyzed against H_2_O for 3 days in the dark, with water being replaced every day [31].

The conjugation was verified by ^1^H-NMR Bruker Avance III-400 MHz, 9.4 T (number of scans 64 and repetition time 6.5 s).

### 2.6. Preparation and Characterization of PEI-Based Complexes

25 kDa PEI was used to prepare PEI/hsDNA complexes: for this purpose, the polymer was dissolved in 0.1 M acetic acid (pH 4) at 50 °C and filtered through 0.025 µm filters. The same procedure used for the preparation of CS-based complexes was used to prepare and characterize PEI/hsDNA (or siRNA tat/rev) complexes, with N/P ratios of 10 and 20. A fixed volume of a 1 mg/mL hsDNA solution (or 20 µM siRNA tat/rev solution) was added dropwise to a 1 mg/mL PEI solution (Table 2). The mixture was stirred for 15 min, incubated for 30 min at RT and then stored at 4 °C for 24 h. Before use, samples were sonicated for 30 min in an ultrasound bath. Hydrodynamic size and Z-potential of the complexes were measured by DLS.

### 2.7. Liposomes Preparation

Commercial Escort IV^TM^ Transfection Reagent was used to prepare liposomes with hsDNA (or siRNA tat/rev) according to the supplier’s protocol. Both lipid suspensions and hsDNA (or siRNA tat/rev) solutions were diluted with fixed volumes of RPMI 1640 medium and then mixed together. The dimensions of the liposomes were measured by DLS.

### 2.8. Cell Experiments

#### 2.8.1. Cells

The CD4+ C8166 T cell line, particularly susceptible to HIV-1 infection, was used for in vitro experiments. The C8166 human cell line was purchased by the ATCC (Manassas, VA, USA) and kept in the culture at the Virology laboratory at the Department of Molecular Medicine, Sapienza University, Rome, Italy. Cells were maintained in culture at 37 °C in RPMI 1640 medium enriched with 10% FCS (Fetal Calf Serum), 1% glutamine and 0.5% gentamicin and sub-cultivated every 72–96 h in fresh medium.

#### 2.8.2. Cell Viability

Cells were seeded into 96-well plates at 3 × 10^4^ cells/well in a complete medium and treated with different concentrations of CS particles: 100 µg/mL, 20 µg/mL and 10 µg/mL at incubation times of 24, 48 and 72 h. At each prefixed time, 80 µL of the medium was replaced with 20 µL of MTT solution (5 mg/mL in PBS). After 2 h incubation at 37 °C in the dark, isopropyl alcohol/Triton X/HCl solution was added to allow cellular lysis, and the absorbance at 570 nm was determined spectrophotometrically. The viability was calculated using the following equation:$$Viability=\frac{Absorbance\;of\;treated\;cells}{Absorbance\;of\;control\;cells}\times 100$$

#### 2.8.3. Cellular Uptake of CS/hsDNA Complexes

The uptake of FITC-labelled CS particles by C8166 was studied with a fluorescence-activated cell sorting analysis (MACSQuant^®^ Analyser, Miltenyi Biotec, Bergisch Gladbach, Germany). Particles suspension was diluted in RPMI serum-free medium at a 0.1 mg/mL concentration and was then incubated for 15 min at room temperature.

In double-click tubes, 106 cells were seeded and suspended in 2 mL RPMI serum-free medium with 500 µL of NPs suspension. Samples were incubated for 4, 6 and 24 h using particles concentrations of 100, 20 and 10 µg/mL.

After incubation, control cells and cells treated with FITC-labelled CS NPs were washed with PBS, centrifuged (1800 rpm for 10 min), resuspended in 300 µL of PBS and finally analyzed by FACS (fluorescence-activated cell sorting) to determine the percentage of FITC+ cells (stop gate: 100,000 cells). MACSQuant^®^ Calibration Beads (Miltenyi Biotec) were used for the calibration of the instrument. The analysis of the acquisitions was developed using the MACSQuantify^®^ software version 2.5 (Miltenyi Biotec) with the same gating strategy applied to treated and untreated samples. The use of untreated control cells allowed uniquely identifying the fluorescence of the cell population under analysis.

FACS analysis was also performed on C8166 cell lines infected with HIV-1 to investigate nanocomplexes uptake in the presence of HIV-1 virus.

#### 2.8.4. Cell Infection

Cells were infected with the HIV-1 P1 virus [32] with a multiplicity of infection (MOI) of 1 TCID50/cell. Cells were incubated with the virus for 1 h at 37 °C, and after 24 h, the HIV-1 cytopathic effect was observed.

#### 2.8.5. Gene Transfection and Antiviral Activity

CS-based complexes loaded with hsDNA or siRNA tat/rev were used for transfection experiments, and their antiviral activity against HIV-1 was compared with PEI particles and ESCORT IV liposomes. Cells were treated with each of the above-mentioned preparations before or after infection with 4 h of incubation. A single nanocomplexes dilution of 20 µg/mL was chosen. Antiviral activity was evaluated by measuring HIV RNA in the supernatant by the VERSANT^®^ kPCR Molecular System (Siemens Healthineers, Erlangen, Germany), and the percentage reduction in viral replication was calculated as follows:$$Red\%=\frac{RNA\;in\;control\;sample-RNA\;in\;particles\;solution}{RNA\;in\;control\;sample}\times 100$$

## 3. Results and Discussion

### 3.1. Preparation and Characterization of CS/hsDNA Complexes

It is well known that particle size and size distribution influence different properties of micro- and nanoparticles, from their interaction with biological systems to entrapment stability and nucleic acid release kinetics [33]. In order to study such phenomena, different CS/hsDNA samples were initially prepared using CS of both 50 and 150 KDa and three different N/P ratios (5, 10 and 20) in order to optimize parameters such as particle dimensions and entrapment efficiency. Hydrodynamic diameter measurements highlight the larger size of the 50 KDa CS-based particles and show increasing values as the N/P ratio increases, probably due to a major chain entanglement (Figure 1 and Table 3) [34]. Molecular weight and N/P ratio also influence Z-potential, which is more positive for 50 KDa CS-based complexes and larger N/P values (Table 4), but they do not influence the PdI, whose values are always in the 0.1–0.3 range for all samples, evidence that the different preparations are highly monodispersed.

Particle size is considered a key parameter for the design of drug delivery nano or micro-systems since particle size may affect the ability to overcome the transport barriers in biological tissues [35]. Our data showed that an N/P ratio of 20 afforded the biggest particles for both CS molecular weights. For choosing the parameters to employ in the rest of the work, we also took into consideration the zeta-potential values of the particles since it is well known that this can be used as a quantitative measurement of charge-induced colloidal stability [36]. Zeta-potential values above |30| mV are considered indicative of the electrostatic stability of colloidal particles. Therefore, on such basis, we chose to conduct the rest of the work using the conditions that afforded the preparation of particles with a zeta-potential value above |30| mV (this was achieved only with a 50 kDa chitosan) and, between the two remaining experimental conditions (N/P 10 or N/P 20), we chose the one that afforded particles with lower size (N/P 10). On such a basis, all subsequent experiments were conducted using complexes formulated with a 50 kDa CS and an N/P ratio of 10.

Field emission scanning electron microscopy (FE-SEM) analysis was performed to examine the morphology of the complexes. Figure 2 shows a particle population uniform in size, with spherical morphology and an average diameter of 173 nm. Due to the high hydration of complexes, as expected, there is a discordance between this value and that of the hydrodynamic diameter obtained by DLS.

The entrapment efficiency, measured by the spectrophotometric method described in Section 2.2, was greater than 80% (81.3 ± 0.6%), which demonstrates the high binding affinity between CS and hsDNA. In order to confirm the entrapment of hsDNA within chitosan particles, an electrophoretic gel analysis was carried out. As can be seen from Figure 3, the fluorescence given by ethidium bromide appears mostly inside the wells. This confirms the entrapment of hsDNA within chitosan particles, excluding its adsorption on the surface.

In vitro release kinetics studies were performed for 80 h (Figure 4). Generally, in micro- or nanoparticle release systems, a burst effect is often observed due to the load fraction adsorbed on the surface of the complexes or weakly linked to the polymeric matrix. Instead, the analyzed complexes showed a quite limited burst effect: in the first 4 h of the study, DNA release was not greater than 11%, and the curve reached a plateau value after 48 h, corresponding to a quantity of released hsDNA equal to 40% of the total loaded amount. hsDNA in vitro release profile suggests a slow and sustained diffusion mechanism of hsDNA from complexes due to a very stable interaction between CS and the nucleic acid.

### 3.2. Preparation and Characterization of FITC-Labelled CS

In order to study the in vitro cellular uptake of CS-based complexes, CS molecules were chemically functionalized with a fluorescent probe, fluorescein isothiocyanate (FITC) isomer I, through covalent binding.

The UV-Vis absorption spectra of FITC-CS in acetic acid (pH 4) are characterized by a peak at 456 nm, followed by a second peak of lower intensity at 486 nm, characteristic of FITC. As expected, no absorption is detected for the unfunctionalized polymer. The conjugation efficiency is greater than 70%. In order to exclude any physical adsorption of FITC on CS, a ^1^H-NMR analysis was also performed.

By comparing the CS-FITC ^1^H-NMR spectrum (Figure 5B) with the CS ^1^H-NMR spectrum (Figure 5A) and with FITC spectra present in the literature [37,38], the presence of their characteristic peaks can be observed in CS-FITC spectrum (related to CS carbohydrate rings and acetyl groups and FITC aromatic rings at 6.7ppm). It was calculated, through the ratio between the C2 signal area of CS and the sum of the areas of all the six aromatic protons of fluorescein, that about 3.55% of the rings of the CS chain are covalently linked to FITC.

### 3.3. Preparation and Characterization of CS/siRNA Tat/Rev Complexes and Control Carrier Systems for Cellular Transfection

After studying CS/hsDNA as a model system, CS complexes for siRNA tat/rev HIV-1 specific cell transfection were prepared through ionotropic gelation.

Moreover, in this case, the hydrodynamic size and Z-potential of the complexes were characterized by DLS measurements, and the entrapment efficiency was calculated through data obtained by UV-vis spectrophotometry (Figure 6a and Table 5). Data show a particle population with a smaller diameter, more polydisperse and with a higher Z-potential than the CS/hsDNA model system. The entrapment efficiency, on the other hand, is comparable to the value obtained for CS/hsDNA complexes.

### 3.4. Preparation and Characterization of PEI and Escort IV-Based Complexes

Two different particle systems were used in addition to CS for synthetizing microvectors for transfection experiments of siRNA tat/rev: (1) polyethylenimine (PEI), a linear, cationic polymer, and (2) Escort IV^TM^ Reagent to prepare uncharged liposomes. A chemico-physical characterization was also performed for both model systems loaded with hsDNA. Table 6 shows the high entrapment efficiency of both systems used, while Figure 6b,c report the size distribution of the complexes obtained by DLS measurements.

### 3.5. Cell Viability and Uptake of Complexes

In order to estimate the potentially toxic effects of CS on cell viability, an MTT assay was performed on the C8166 cell line to assess cellular metabolic activity in the presence of complexes.

Three different dilutions (100, 20 × 10^10^ µg/mL) and three different incubation times (24, 48 and 72 h) were tested, and the viability percentage was calculated. The results reported in Figure 7 clearly show that the cellular metabolic activity was not affected by the presence of CS complexes (*p* = 0.9; chi-square test).

Nevertheless, optical microscopy analysis (Figure 8) highlighted a morphological alteration for the C8166 cell line when treated with high sample concentrations: typical cell clusters disappear and are replaced by disseminated and isolated cell growth (Figure 8).

In order to determine the complexes’ ability to cross the cell membrane and be efficiently internalized by cells, cellular uptake studies by FACS analysis were carried out using complexes prepared with FITC-labelled CS entrapping hsDNA at three different concentrations and three different times on C8166 cell line.

A quantitative measure of FITC+ cells, i.e., the percentage of fluorescent cells due to the internalized labeled complexes, was obtained (Figure 9), indicating, for both cell lines, that the endocytosis process is independent of the incubation time. Endocytosis is very efficient, as after 4 h, all the complexes have entered cells.

On the basis of the obtained results on cell viability and uptake, for all subsequent experiments on infected cells, complexes with a 20 µg/mL concentration and an incubation time of 4 h were used in order to allow high uptake without obtaining a change in the morphology of the C8166 cell line.

Data from FACS analysis on infected cells indicate how HIV infection could influence particle uptake (Table 7). In fact, C8166 cells showed a remarkable decrease in particle uptake in both conditions of infection (before or after the administration of CS complexes) in comparison with non-infected cells (Table 7). HIV-1 is known to modify membrane permeability through the insertion of transmembrane proteins (viroporins) that alter intracellular Na^+^ concentration, with a consequent pH decrease and water recall that could alter the uptake of the particles [39,40]. Furthermore, viroporins could interfere with intracellular traffic and promote complex escape.

### 3.6. Antiviral Activity of CS-siRNA Complexes

After the development of a model CS-based delivery system, the antiviral activity of CS/siRNA tat/rev complexes was tested only on HIV-1 infected cells by measuring HIV-RNA in the culture medium.

Figure 10 reports the results of the viral inhibition experiments by using the different selected vectors for the transfection, in two different modes (pre-infection in Figure 10A or post-infection in Figure 10B), of the human cell line C81666 infected with HIV-1 virus. We would like to point out that, on the basis of the experimental conditions employed, the post-infection conditions can be considered the most efficient approach to evaluate the in vitro effect of siRNA delivery by using polymeric microvectors. By comparing the infection reduction ability of the different vectors (in the absence of SiRNA molecules), we can conclude that CS is the only polymer able to induce a significant cell inhibition in pre- as well as post-infection conditions. On this basis, we can point out that, as is generally assessed, CS is recognized to improve anti-infection actions, and many CS-based systems are employed for the treatment of infectious diseases mediated by bacteria and viruses [41].

Moreover, while PEI-based vectors did not significantly affect viral replication, CS-based vectors showed a comparable ability to inhibit virus replication with those based on Escort.

In particular, when using CS-based vectors, it seems that both using CS/hsDNA model system and CS/siRNA in transfection experiments in the pre-infection mode, we observed a lower viral inhibition (up to 40%). We think that in this case, particles entrapped into cells before infection have partially released the siRNA, which may have been enzymatically degraded.

On the contrary, when using the same complexes in post-infection transfection mode (i.e., treating cells with CS particles after viral infection), higher viral inhibitions were achieved (from 60 to 70%). In the literature, the antiviral and antimicrobial activity of CS is already known, and many authors have evidenced the ability of CS molecules to favor electrostatic interactions with retroviral surface proteins preventing the fusion of different viruses with cell membranes [42,43,44]. In this case, we can hypothesize that CS-based particles used to treat infected cells may be able to electrostatically complex virions, thus contributing to increased viral inhibition.

These results are also confirmed by those obtained when using PEI complexes for transfection. In fact, these vectors caused a significant HIV-1 inhibition (30%; *p* = 0.015) only with post-infection treatments (Figure 10). Different phenomena can be taken into account when considering the transfection of C8166 cells with Escort IV liposome-based vectors. In this case, the liposome-based vectors entrapping model hsDNA are probably not able to inhibit virions in pre- or post-infection treatments (lipid membranes do not complex viral particles), but when using siRNA complexed liposome vectors, we obtained in both pre- and post-infection conditions significant viral inhibitions (*p* ≤ 0.0001) due to an increased ability of such model particles to cross cell membranes (Figure 10). However, liposomes have several disadvantages as vectors for drug delivery as they may be subjected to oxidative degradation and have high production costs [45].

## 4. Conclusions

This work investigated the antiviral activity of CS/siRNA tat/rev complexes in the in vitro HIV targeting experiments. As generally recognized, the main problems in the development of siRNA-based drugs for therapeutic use are the low efficiency of siRNA delivery to target human cells infected by viruses. One of the most promising approaches to solve the problem is bioconjugation with biopolymers. We chose positively charged polymers as chitosan of medium MW to entrap and efficiently deliver siRNAs targeting viral proteins within cells infected by HIV. Moreover, this study provided interesting results on the effects on HIV-1 virus inactivation by treating infected cells in “post-infection mode“, thus achieving high viral inhibition. This gene silencing is considered a good approach for in vitro silencing, despite the fact that frequent mutations in the HIV-1 genome could be able to abolish its effectiveness. Thus, in the future, a cocktail of siRNAs targeting different HIV-1 gene sequences would be more likely to avoid HIV-1 mutant escape.

## Figures and Tables

**Figure 1 materials-15-05340-f001:**
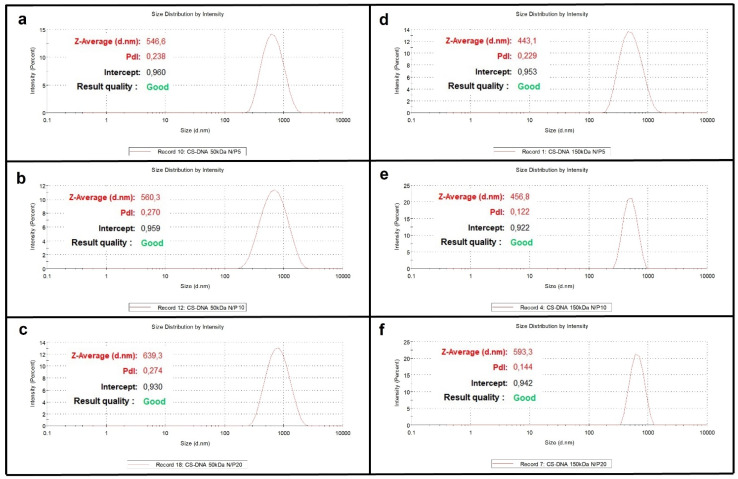
Size distribution by intensity of CS/hsDNA complexes prepared with CS of different molecular weights and at different N/P values (from N/P 5 to N/P20 by using 50 and 150KDa chitosan).

**Figure 2 materials-15-05340-f002:**
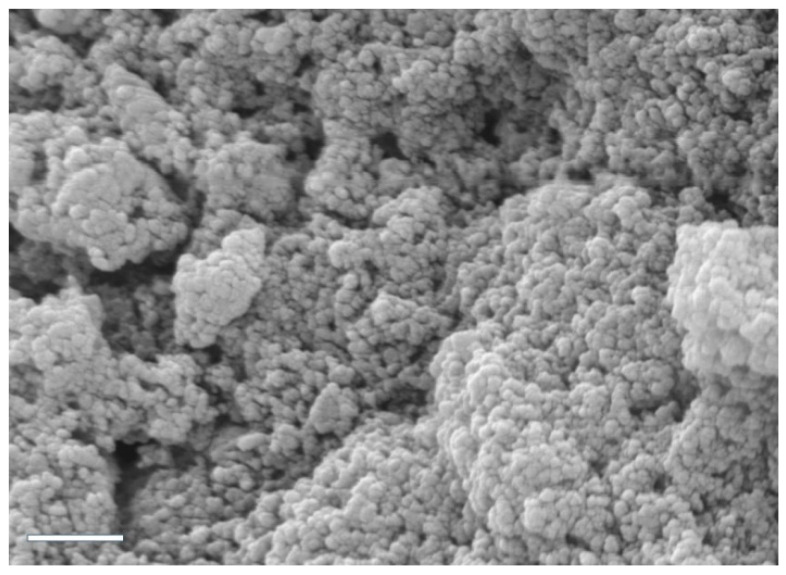
SEM micrograph of CS/hsDNA particles (CS molecular weight = 50 kDa, N/P ratio = 10). Scale bar: 2 µm.

**Figure 3 materials-15-05340-f003:**
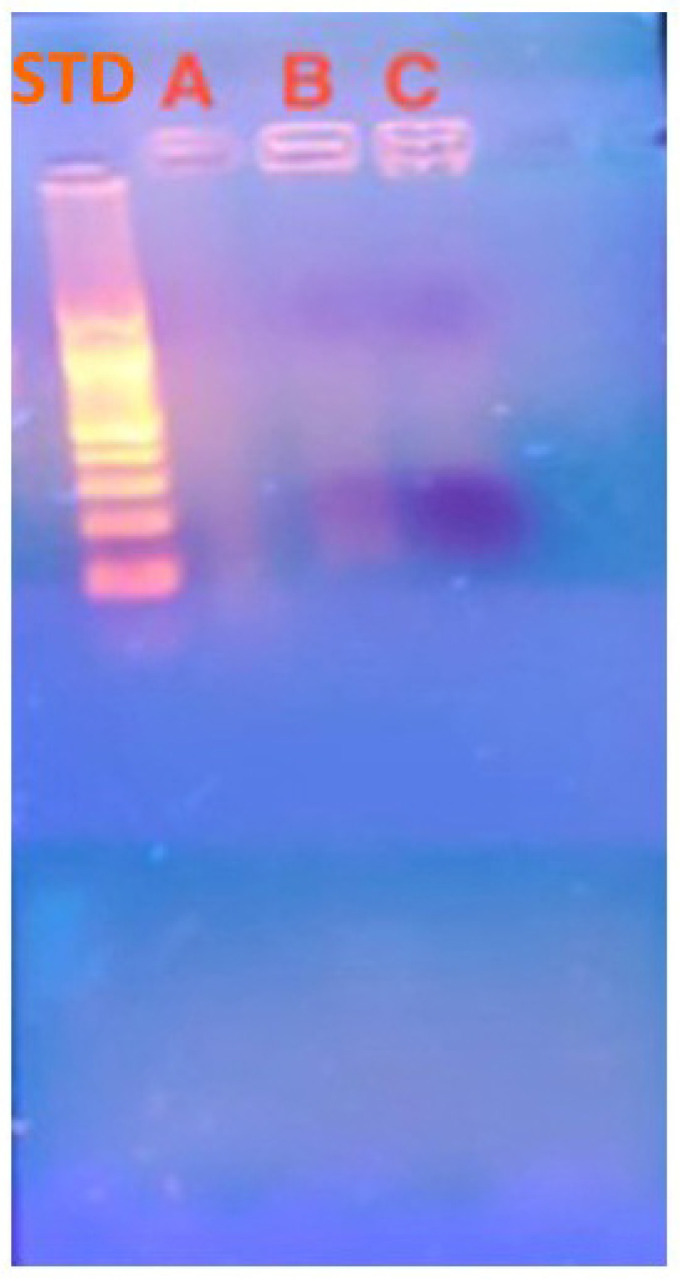
Two percent Agarose gel electrophoresis: (STD) DNA standards of different molecular weight, (A) CS/hsDNA microparticles, (B) CS/hsDNA in the presence of DNAse I, (C) CS/hsDNA in the presence of inactivated DNAse I.

**Figure 4 materials-15-05340-f004:**
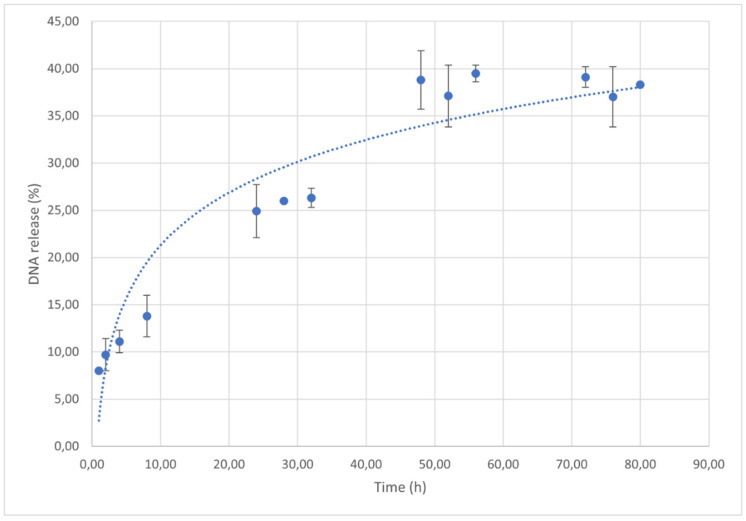
In vitro release kinetics of hsDNA from CS/hsDNA complexes. N/P ratio = 10, CS MW = 50 kDa.

**Figure 5 materials-15-05340-f005:**
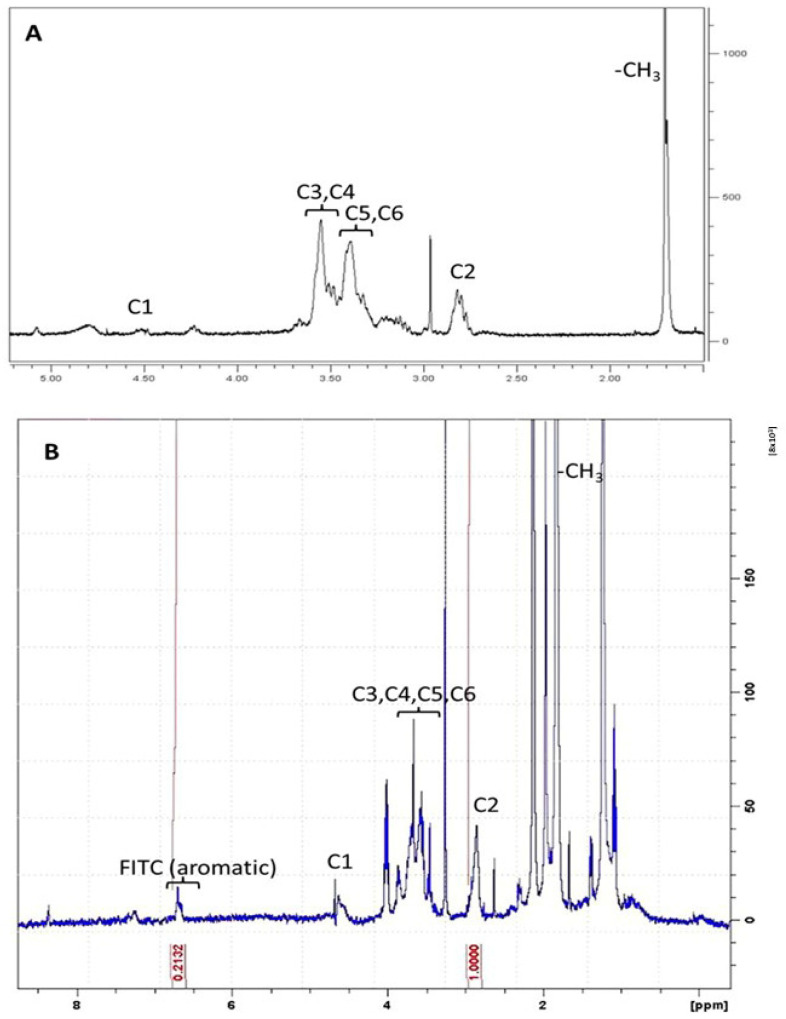
^1^H-NMR spectra of CS (**A**) and CS-FITC conjugate (**B**) in D_2_O acidified with DCl. The signals relating to CS (black line in box (**A**)) and CS FITC (blu line in box (**B**)) and the integrals (black lines in box (**B**)) relating to the diagnostic signals of the two species are reported.

**Figure 6 materials-15-05340-f006:**
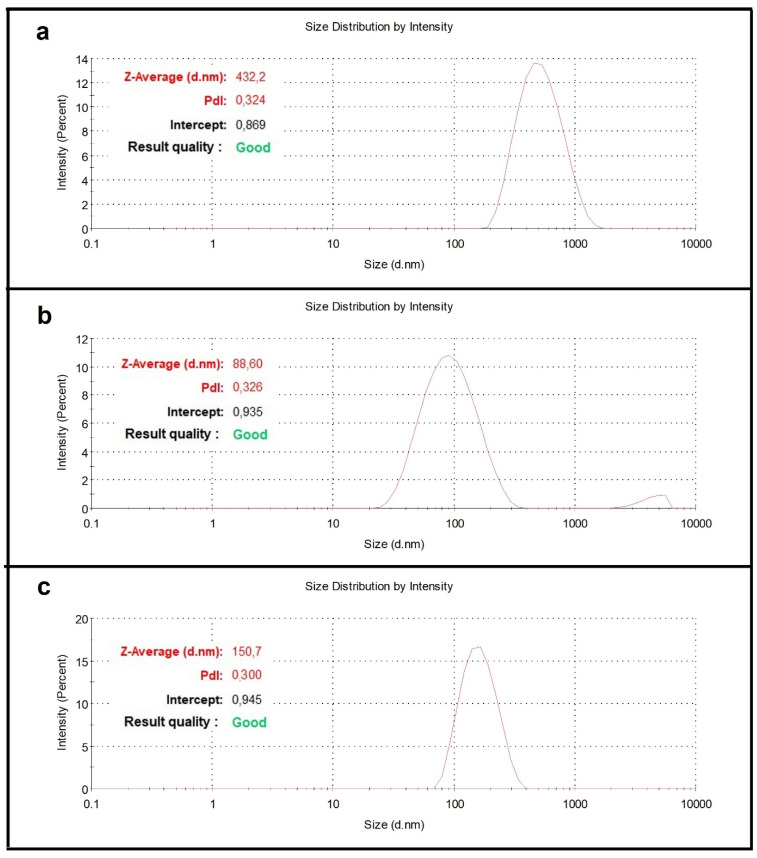
Size distribution by intensity of CS/siRNA (**a**), PEI/siRNA (**b**) and Escort IV^TM^/siRNA (**c**) complexes.

**Figure 7 materials-15-05340-f007:**
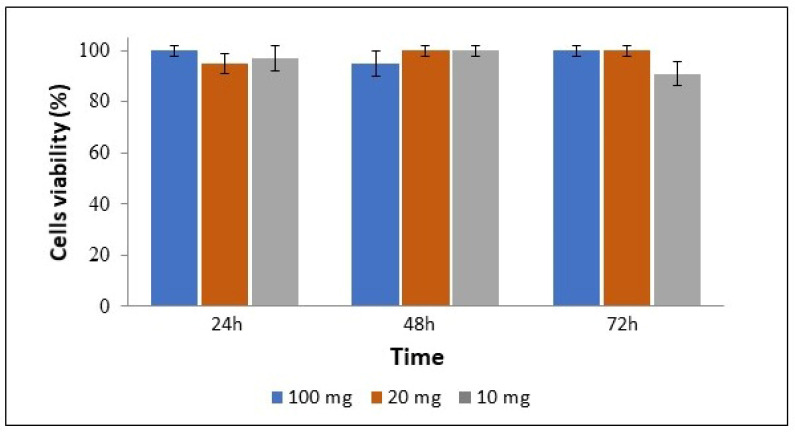
Cell viability of C8166 cells after treatment with CS/hsDNA complexes at concentrations of 100, 20 × 10^10^ µg/mL for 24, 48 and 72 h.

**Figure 8 materials-15-05340-f008:**
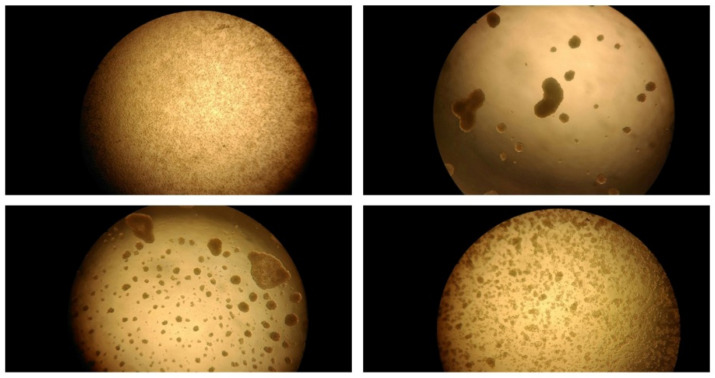
Optical microscope image of C8166 cells treated with different concentrations of CS/hsDNA complexes (CS MW = 50 KDa, N/P = 10): 250, 50, 10 and 0 µg/mL.

**Figure 9 materials-15-05340-f009:**
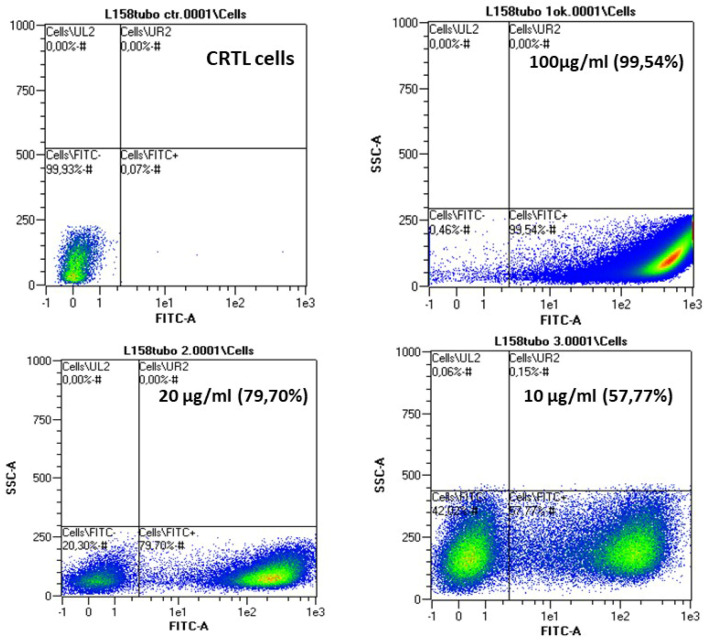
Fluorescence-activated cell sorting of C8166 treated with 0, 100, 20 and 10 µg/mL of FITC-CS/hsDNA particles for 4 h of incubation at RT (CS MW = 50 KDa, N/P = 10).

**Figure 10 materials-15-05340-f010:**
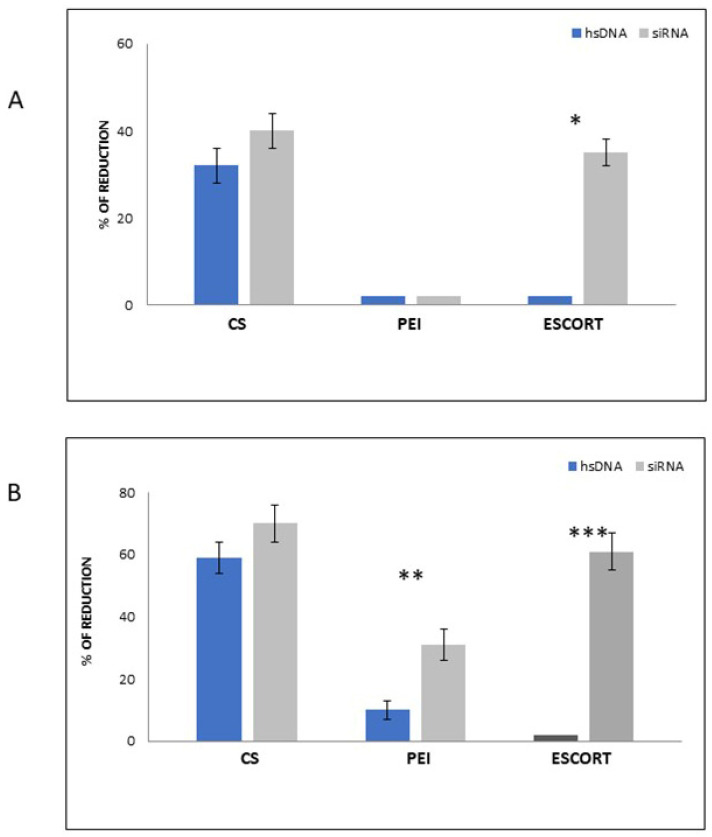
Percentage of HIV-1 RNA reduction in C8166 cell line treated for 4 h pre (**A**) and post (**B**) single cycle HIV-1 infection, by PCR analysis. Both cell lines were treated with each nano delivery system (CS and control systems) loaded with both hsDNA and siRNA tat/rev at N/P = 10, with 20 µg/mL. * *p* = 0.0015; ** *p* = 0.0001; *** *p* < 0.0001 (Chi-square test).

**Table 1 materials-15-05340-t001:** Weight ratios (*w*/*w*) between CS and nucleic acid used to synthetize polymeric particles with different N/P values.

N/P	CS (mg)	HsDNA or siRNA (µg)
5	1	320
10	1	160
20	1	80

**Table 2 materials-15-05340-t002:** Weight ratios (*w*/*w*) between PEI and nucleic acid used to synthetize polymeric particles with different N/P values.

N/P	PEI (mg)	hsDNA or siRNA (µg)
10	214	160
20	214	80

**Table 3 materials-15-05340-t003:** Hydrodynamic diameter (nm) of CS/hsDNA complexes prepared with CS of different molecular weights and at different N/P values.

MW (kDa)	N/P 5	N/P 10	N/P 20
50	546.6 nm	560.3 nm	639.3 nm
150	443.1 nm	456.8 nm	593.3 nm

**Table 4 materials-15-05340-t004:** Z-potential (mV) of CS/hsDNA complexes prepared with CS of different molecular weights and at different N/P values.

MW (kDa)	N/P 5	N/P 10	N/P 20
50	+16.0	+35.0	+48.0
150	+10.0	+12.0	+16.0

**Table 5 materials-15-05340-t005:** Hydrodynamic diameter, PdI, Z-potential and entrapment efficiency (EE) of CS/siRNA tat/rev complexes prepared with CS 50 KDa and N/P = 10.

Ø (nm)	PdI	Z-Potential (mV)	EE (%)
432.2	0.3	+40.2	79.9 ± 4.8

**Table 6 materials-15-05340-t006:** Hydrodynamic diameter, PdI, Z-potential and entrapment efficiency (EE) of PEI/siRNA and Escort/siRNA.

	Ø (nm)	PdI	Z-Potential (mV)	EE (%)
**PEI/siRNA**	88.7	0.2	+38	93.0 ± 3.2
**Escort/siRNA**	150.2	0.3	0	95.0 ± 2.8

**Table 7 materials-15-05340-t007:** Percentage of FITC+ cells treated with FITC-CS/hsDNA particles. C8166 cells were treated for 4 h before or after a single cycle of infection with HIV-1.

	Pre-Infection	Post-Infection	Control
**C8166 FITC+**	46.1%	63.3%	75.5%

## Data Availability

Data are contained within the article.

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
