# Peer review of "siRNA Transfection Mediated by Chitosan Microparticles for the Treatment of HIV-1 Infection of Human Cell Lines"

_materials, 2022, doi:10.3390/ma15155340_

Round 1

Reviewer 1 Report

I enjoy reading the work "siRNA transfection mediated by Chitosan microparticles for the treatment of HIV-1 infection of human cell line" by Chronopoulou, et al. The team preseanted a chitosan microparticle formulation for the encapsulation and intracellular delivery of hsDNA and siRNA to silence HIV-1 infection. Although the application proposed is innovative, the results presented lack of rigor. Below I provide major revisions to be addressed:

1. Are the encapsulation and release experiments with CS particles done all in acetic acid? Can the authors comment on the stability of the plasmids at this acidic environment as a function of time? When the plasmids at pH 4 denature?

2. In page 5, sentence in line 187: Can the authors elaborate more in the discussion of ZP and chitosan MW relatioship? One could expect that higher MW polymers contain more amine groups available to complex with plasmids and should be reflected in the highly positive ZP.

3. PDI is very important in the preparation of micro/nano plasmid formulations. The authors must present size data as size distribution plots, including mean hydrodynamic size, standard deviation and PDI. 

4. The authors claim throughout the manuscript "colloidal stability", however, this has not been proven with the data provided currently. Hydrodynamic size distributions must be shown in protein-rich fluids.

5. The SEM images provided do not show clearly morphology nor size of the microparticles. TEM micrographs will be more appropriate to show narrow size distribution of the particles. Additionally, showing all the formulations micrographs will be more compeiling.

6. Since different formulations were originally prepared, they most ideal approach to choose the best formulation will be to evaluate entrapment efficiency and entrapment capacity rather than size and ZP.

7. Entrapment results must be showed. Gel electrophoresis images, etc.

8. How do the authors arrive to the conclusion that a small fraction of plasmid was absorbed on the surface (page 6 line 207)? The authors must elaborate and backup this claim.

9. Giving that chitosan proparties vary from source to source and batch to batch, the authors should run their own CS 1H-NMR and compare it in the same plot with the FITC conjugated one, which will lead to a more accurate calculation.

10. It is unclear why section 3.4. is entitled "In vitro inhibition test of HIV-1 replication", when the section only shows characterization of other formulations.

11. For cell viability results, the authors need to calculate statistical significance.

12. Can the authors provide better images for Figure 5 in which single cell can be distinguished? The current images provided are imposible to judge as cells or cell clusters.

13. The quality of flow cytometry plots needs to be improved.

14. For the HIV-1 RNA reduction percentage results, bar plots will be a more appropiate form of presentations, and statistical significances must be included.

15. Is the % of HIV-1 RNA reduction evaluated in the same size of population for each group?

Author Response

please find the responses in the attachment

Reviewer 2 Report

No explanation of why C8166 human cell was used in the study 

No solid reason of choosing 50 kDa CS-based complexes with N/P ratio of 10 .. Table 3 and 4 displayed numbers for hydrodynamic size and z-potential but no reason for the selection of 560 nm (Also no unit in the table) and +35 for z-potential 

How the entrapment efficiency of 80% was measured , no result is described in page 6. 

How release study was performed and analysed? again no clear explanation 

NMR spectrum should be labelled for each peak 

Author Response

(The authors gave the same response as above.)

Round 2

Reviewer 1 Report

The authors have addressed the majority of my initial concerns. I still belive that the quality of the figures could be improved.

Author Response

Following the reviewer suggestions we have added the new figures with improved resolution (600 dpi)